# Sex-Specific Outcomes in Patients with Acute Coronary Syndrome

**DOI:** 10.3390/jcm9072124

**Published:** 2020-07-06

**Authors:** Johannes T. Neumann, Alina Goßling, Nils A. Sörensen, Stefan Blankenberg, Christina Magnussen, Dirk Westermann

**Affiliations:** 1Department of Cardiology, University Heart and Vascular Center Hamburg, 20246 Hamburg, Germany; a.gossling@uke.de (A.G.); n.soerensen@uke.de (N.A.S.); s.blankenberg@uke.de (S.B.); c.magnussen@uke.de (C.M.); d.westermann@uke.de (D.W.); 2German Center for Cardiovascular Research (DZHK), Partner Site Hamburg/Kiel/Luebeck, Germany; 3Department of Epidemiology and Preventative Medicine, Monash University, Melbourne VIC 3004, Australia

**Keywords:** acute coronary syndrome, sex, myocardial infarction, ST-elevation myocardial infarction (STEMI), non-STEMI (NSTEMI), unstable angina pectoris, outcome, inhospital mortality, trend, Germany

## Abstract

Sex differences in patients with acute coronary syndrome (ACS) are a matter of debate. We investigated sex-specific differences in the incidence, outcomes, and related interventions in patients diagnosed with ACS in Germany over the past decade. All ACS cases from 2005 to 2015 were collected. Procedures and inhospital mortality were assessed by sex. Age-adjusted incidence rates were calculated. In total, 1,366,045 females and 2,431,501 males presenting with ACS were recorded. Females were older than males (73.1 vs. 66.4 years of age), had a longer mean hospital stay (7.7 vs. 6.9 days), and less frequently underwent coronary angiographies (55% vs. 66%) and coronary interventions (35% vs. 47%). The age-adjusted incidence rate of ACS was lower in females than in males, and decreased in both sexes from 2005 to 2015. The age-adjusted inhospital mortality rate was substantially higher in females than in males, but decreased in both sexes over time (in females, from 87 to 71 cases per 1000 person years; in males, from 57 to 51 cases per 1000 person years). In conclusion, we reported sex differences in the incidence, treatment, and outcomes of ACS patients in Germany within the past decade. Women had a substantially higher mortality rate and lower rate of coronary interventions.

## 1. Introduction

Every year, more than 20 million patients present to emergency departments in Europe and North America with suspected acute coronary syndrome (ACS) [1,2]. This makes ACS a highly significant disease to consider in acute cardiac care. Evaluating temporal changes in ACS incidence is important, as is monitoring outcomes in different patient populations.

In recent years, there have been substantial improvements in diagnostic and therapeutic options for patients with ACS, resulting in improved outcomes [3]. These improvements include the evolution of high-sensitivity troponin assays that allow for faster and more accurate diagnostic pathways [4,5]. Primary and secondary prevention strategies have improved over time, and more sophisticated invasive strategies for treating ACS patients are now available. These include the clinical application of routine coronary angiography with the possible use of modern stents, more efficient antiplatelet-therapy regimes, and the more widespread use of lipid-lowering treatments [3]. As a result, several studies documented a decrease in ACS incidence and, an improvement in outcomes [6,7,8]. However, there is an ongoing discussion about sex differences in both treatment and outcomes. Most observed differences were to the detriment of women, who were reported to have a higher rate of misdiagnosis and more delayed diagnoses due to atypical symptoms, and they underwent reperfusion therapy less frequently [9]. In addition to lower resource utilization among females [10,11,12,13], earlier studies reported worse outcomes in women, including a higher incidence of ACS-related mortality [11].

Currently, there are limited data from sex-specific analyses on large contemporary datasets. Therefore, we aimed to assess differences in incidence, outcomes, and related interventions in all females and males hospitalized for ACS in Germany over the past decade.

## 2. Methods

### 2.1. Study Design

For this project, data from the Federal Bureau of Statistics in Germany were used. In this institution, all clinical cases in Germany are collected in a central database according to their International Classification of Diseases (ICD), and Operationen und Prozedurenschlüssel (Operations and Procedures Classification; OPS) codes. Data were exported on our behalf by the Research Data Center of the Federal Statistical Office and Statistical Offices of the Federal States in Wiesbaden, Germany, and aggregated statistics were provided on the basis of R codes that we supplied to the Research Data Center. We investigated all cases of patients with a main diagnosis of ACS (ICD Codes I20.0, I21 or I22) in Germany from 2005 to 2015.

There was no commercial support for the study or the preparation of this manuscript. This study did not involve the researchers gaining direct access to data on individual patients—only to completely anonymized summary results provided by the Research Data Center. Therefore, approval by an ethics committee and informed consent were not required, in accordance with German law.

### 2.2. Diagnoses and Procedural Codes

In Germany, it is mandatory to transfer data to the Institute for Medical Documentation and Information (Cologne, Germany) on diagnoses, coexisting conditions, and procedures for every patient to enable reimbursement under the hospital remuneration system. Coding guidelines and annual adaptations ensure uniform documentation. Diagnoses and procedures are coded according to the international statistical classification of diseases and related health problems, 10th revision, German modification (ICD-10-GM), and Germany’s procedure classification system, the OPS. A list of ICD and OPS codes that were used for the present analysis is provided in Appendix A. Briefly, the characteristics examined in our research were age, sex, hypertension, dyslipidemia, diabetes, peripheral artery disease (PAD), and a history of stroke, atrial fibrillation, chronic obstructive pulmonary disease (COPD), renal insufficiency, and coronary-artery bypass grafting (CABG). The studied inhospital procedures were coronary angiography, percutaneous coronary intervention (PCI), and CABG. Furthermore, information on inhospital mortality was collected.

### 2.3. Statistical Analysis

Binary variables were shown as absolute numbers and percentages, and were compared to the ***χ***^2^ test, whereas continuous variables were shown as means ± standard derivation (SD), and were compared to one-way analysis of variance (ANOVA). Moreover, age-adjusted incidence rates were calculated for ACS, ST-elevation myocardial infarction (STEMI), non-ST-elevation myocardial infarction (NSTEMI), and unstable angina pectoris (UAP) using Germany’s age distribution on 31 December of each year as a reference. ACS distribution in Germany was applied as a reference for the age-adjusted incidence rates of inhospital mortality. Additional analyses were performed according to the type of stent used during PCI, and for patients with STEMI, NSTEMI, and UAP. All statistical methods were written in R statistical software, version 3.5.2 (R Project for Statistical Computing, https://www.r-project.org), and performed at the Federal Statistical Office in Germany.

## 3. Results

### 3.1. Study-Population Characteristics

In the overall dataset, 1,366,045 females and 2,431,501 males presenting with ACS were recorded (see Table 1). Females were older than males, with a mean age of 73.1 ± 12.2 years in females and 66.4 ± 12.5 years in males. Hypertension was diagnosed in 57.8% of females and 57.1% of males, dyslipidemia in 39.2% of females and 45.9% of males, and diabetes in 29.8% of females and 25.7% of males. There were 299,305 cases of STEMI, 501,080 cases of NSTEMI, and 525,143 cases of UAP in females. In males, 626,199 cases of STEMI, 830,434 cases of NSTEMI, and 916,733 cases of UAP were recorded. In both sexes, patients diagnosed with NSTEMI were older and had a worse cardiovascular-risk-factor profile than that of STEMI patients.

Mean hospital stay was 7.7 ± 7.8 days for females, and only 6.9 ± 7.6 days for males. Patients diagnosed with myocardial infarction (MI) stayed longer than patients with UAP.

Among females, 55.2% received coronary angiography, and 34.6% received coronary intervention. These percentages were higher in males, among whom 66.2% received coronary angiography and 46.7% received coronary intervention. Patients diagnosed with STEMI were more likely to receive angiography or intervention; interestingly, the rates of performed procedures were similar between patients diagnosed with NSTEMI and those diagnosed with UAP. CABG was performed on 3.2% of all females and 5.6% of all males.

In the overall dataset, more patients received a drug-eluting stent (DES) than a bare-metal stent (BMS; see Appendix A), irrespective of sex. Patients who received a DES were younger, and had a shorter hospital stay and a lower inhospital mortality rate than patients who received a BMS.

### 3.2. Temporal Trends from 2005 to 2015

From 2005 to 2015, patients diagnosed with ACS increased in age, a trend that was more prominent in males (mean age: 65.6 vs. 67.0 years in 2005 and 2015, respectively) than in females (mean age: 72.7 vs. 73.2 years in 2005 and 2015, respectively; see Table 2, Appendix A). The prevalence of cardiovascular risk factors increased slightly in both sexes, while the mean duration of hospital stays decreased from 8.4 to 6.8 days in females, and from 7.4 to 6.4 days in males. The rates of coronary angiographies and coronary interventions increased from 44.9% and 26.4% to 64.5% and 42.7%, respectively, in females. A similar trend was seen in males, with an increase in coronary angiographies and coronary interventions from 57.0% and 38.3% to 72.6% and 52.7%, respectively.

The age-adjusted incidence rates of ACS, STEMI, NSTEMI, and UAP were substantially higher in males than in females (see Figure 1 and Appendix A). Within the past decade, ACS incidence rate decreased from 4.26 to 2.96 cases per 1000 person years in females, and from 7.65 to 5.92 cases per 1000 person years in males. In subgroup analyses, the incidence rates of UAP and STEMI also decreased in both sexes, while the incidence rate of NSTEMI increased from 1.01 to 1.34 cases per 1000 person years in females, and even more so in males, from 1.67 to 2.54 cases per 1000 person years.

### 3.3. Inhospital Mortality

Inhospital mortality was substantially higher in females with ACS (7.9%) than in males (5.4%). These findings were confirmed in age-adjusted analysis. In both sexes, inhospital mortality sharply declined from 2005 to 2015. Between 2005 and 2015, the age-adjusted incidence rates decreased from 87.17 to 70.98 cases per 1000 person years in females, and from 57.49 to 51.29 cases per 1000 person years in males (see Figure 2 and Appendix A). In subgroup analyses according to age categories, ACS cases occurred more often in younger males than in younger females. However, the inhospital mortality rate was slightly higher in females than in males (Appendix A).

## 4. Discussion

In a population of more than 1.3 million females and 2.4 million males, we noted a decreasing incidence of acute coronary syndrome in Germany within the past decade. We observed substantial sex differences, with more frequent admissions for ACS in men than in women. Females presented at a higher age than men did and were less likely to undergo coronary angiography or coronary intervention. Inhospital mortality was higher in females than in males (see Figure 3). Because of the contemporary nature of the data and the large sample size, our study added to the existing body of evidence on clinically relevant sex differences in ACS patients.

Both the recent literature and our study reported a decrease in ACS incidence over time. By contrast, the proportion of NSTEMI cases increased. The reasons for this increase could be the better detection of even small-scale myocardial injury using high-sensitivity troponin assays, and an aging population with more frequent comorbidities [4,14]. In recent decades, sex differences in patients with myocardial infarction were discussed frequently [10,11,12,13]. Others and we found a higher rate of ACS in men than in women [15]. This could be related to a higher cardiovascular-risk-factor profile in men [16]. In our study, risk factors were differentially distributed between sexes, with a higher prevalence of dyslipidemia in men being one of the most important risk factors for ACS [17,18].

Despite higher rates of ACS in men, there were substantial sex differences in treatment strategies. We and others [16,19,20] showed that women presenting to the emergency department with chest pain were less likely to undergo invasive diagnostic tests. Although about 90% of patients with ACS present with chest pain or discomfort, some patients do not suffer from typical chest pain [21]. It is well known that women present more frequently with atypical symptoms, such as dyspnea, vomiting, or throat and jaw discomfort [22]. Although sex differences in chest-pain characteristics are small and do not really support an early ACS diagnosis [15], the characterization of chest pain is a precondition for initiating further diagnostic and therapeutic steps [23]. Thus, atypical symptoms may have led to underusing invasive strategies in our female study population. It is also hypothesized that the relative underestimation of pain severity could have resulted in longer delay until diagnosis and treatment [24]. Whether this is directly reflected in sex-specific care in Germany needs to be prospectively tested in future studies.

Vogel et al. showed that fewer females than males underwent an early invasive-treatment approach, and that higher age and renal insufficiency were the strongest predictors for a conservative approach [20]. However, ACS patients with more comorbidities were shown to benefit from invasive treatment strategies [25]. Once invasively diagnosed, females received less revascularization [26], a fact that could be partly explained by differences in coronary status: Isorni et al. found that women much more frequently had normal coronary arteries or nonsignificant coronary-artery disease, irrespective of the ACS type that they had, while men had more severe coronary-artery disease [27]. A higher rate of nonsignificant coronary-artery or microvascular disease could explain the lower PCI rates found in our study, although this could not be proven by our results. A potentially higher prevalence of small-vessel disease in females is also speculated to be a reason for the higher intervention rates in males than in females.

Although global-population aging is linked to a high cardiovascular-disease burden, others and we noted a decline in ACS-related mortality rates in recent decades [25,28,29]. This could be explained by greater adherence to treatment-guideline recommendations. However, outcomes differed by sex, and worse survival rates for females with ACS were previously reported [30,31,32,33]. A decade prior to our study, Vaccarino et al. showed that younger women in particular experienced a larger decrease in ACS-related mortality than men did, mainly because of changes in comorbidity and clinical severity [34]. Although smaller studies with low event rates found no sex differences in inhospital [20] and short-term outcomes [19], long-term mortality was significantly higher in females than in males, but differences diminished after adjustment for age, comorbidities, and treatment [20]. We reported a higher ACS-related mortality rate in women than in men over a very short follow-up period. In prior clinical trials investigating ACS patients, females were often at lower risk for adverse events, a trend that could be explained by certain under-representation [35]. For example, in large pooled analysis of trials conducted by the “thrombolysis in myocardial infarction” (TIMI) study group, only 29% of all participants were females. Thus, our dataset capturing all ACS cases in Germany added important knowledge to combat this possible selection bias. The effects of aging may have influenced our results since others and we demonstrated that women with ACS were older than men with this condition [16,27,36]. By contrast, the recent literature verified that younger women in particular had higher death rates during hospitalization, and that the younger the patient, the higher the risk of death among women relative to that among men [11]. However, although adjustment for age diminished sex differences in ACS-related mortality in other studies [20], our results showing mortality rates that were to women’s detriment remained robust. Differences in presentation alone could not explain the markedly worse outcomes for women, as there is evidence that sex differences in initial symptoms did not translate into sex-specific outcomes in patients with suspected ACS [16]. Differences in therapeutic strategies may have influenced sex-specific mortality. Prior studies suggested that early invasive treatments could significantly reduce inhospital mortality in both women and men [37]. In our large registry, women received markedly fewer invasive procedures, although mortality was shown to be higher in women than in men once obstructive coronary-artery disease was diagnosed [38]. It is well known that a DES strategy is linked to lower coronary-event rates and lower ACS-related mortality [39]. In line with this fact, our analysis showed a lower short-term mortality rate in patients who received a DES than in patients who received a BMS, although these results were similar for both sexes. Therefore, stent type alone could not explain sex differences in inhospital mortality.

Our analysis has its strengths, but it also has limitations. We could collect data on a very large population, encompassing all cases of ACS in Germany within the past decade. However, because the database was a registry, we could not exclude the possibility of reporting bias, as diagnoses and procedures were coded by each hospital without central adjudication. This could have resulted in misclassifications of ACS diagnoses, and the under- or over-reporting of comorbidities. On the other hand, because of the large sample size, this effect was unlikely to have had an impact on the overall results. In addition, our understanding of the observed sex differences could have been improved by knowing possible confounding factors, such as medical treatments, cardiac biomarkers, or other risk factors, including smoking, family history, or socioeconomic factors, which were unfortunately not available in this dataset.

In conclusion, we observed sex differences in the incidence, treatment, and outcomes of patients with acute coronary syndrome in Germany within the past decade. There was a substantially higher mortality rate and a lower rate of coronary interventions in women. This report aimed to raise awareness of these differences in patients, and further efforts are required to overcome these differences.

## Figures and Tables

**Figure 1 jcm-09-02124-f001:**
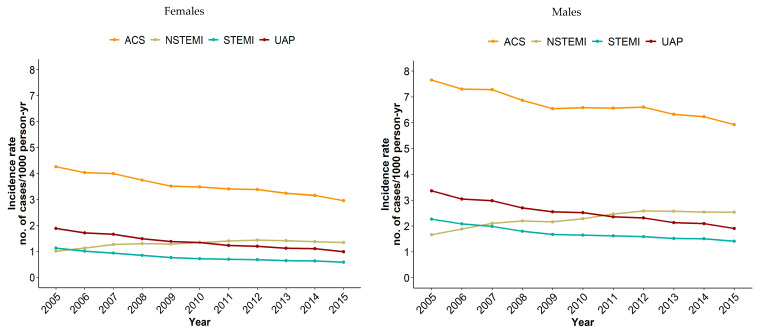
Age-adjusted incidence rates of ACS, NSTEMI, STEMI, and UAP in males and females in cases per 1000 person years from 2005 to 2015.

**Figure 2 jcm-09-02124-f002:**
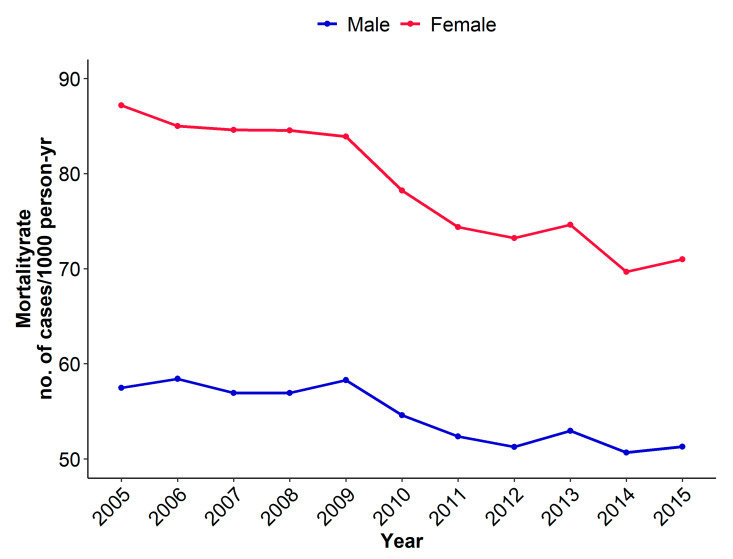
Incidence rates of inhospital mortality in cases per 1000 person years from 2005 to 2015, adjusted for age.

**Figure 3 jcm-09-02124-f003:**
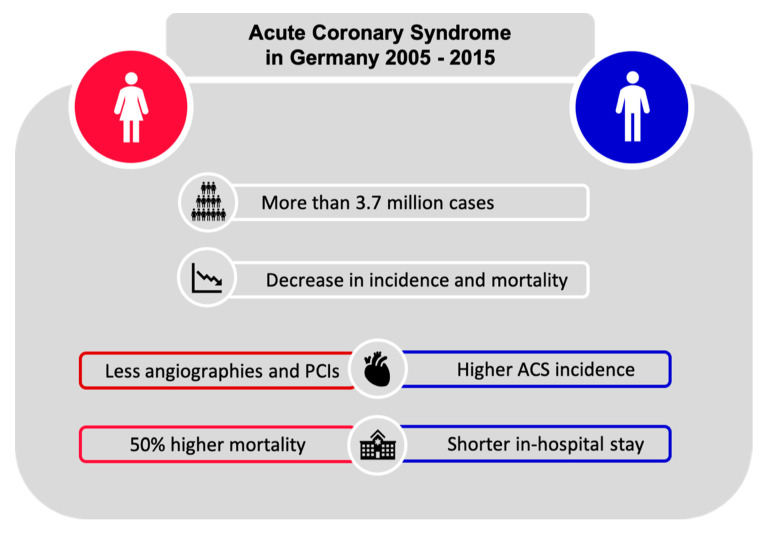
Central illustration summarizing main study results.

**Table 1 jcm-09-02124-t001:** Characteristics of patients with acute coronary syndrome (ACS), and subgroups of unstable angina pectoris (UAP), ST-elevation myocardial infarction (STEMI), and non-ST-elevation myocardial infarction (NSTEMI) in Germany from 2005 to 2015, separated by sex. PAD, peripheral artery disease; COPD, chronic obstructive pulmonary disease; CABG, coronary-artery bypass grafting; PCI, percutaneous coronary intervention.

	Females				Males				*p*-Value Comparing ACS in Females and Males
	ACS	STEMI	NSTEMI	UAP	ACS	STEMI	NSTEMI	UAP	
N Available	1,366,045	299,305	501,080	525,143	2,431,501	626,119	830,434	916,733	<0.001
Age (years)	73.1 ± 12.2	72.2 ± 13.2	75.9 ± 11.5	70.5 ± 11.6	66.4 ± 12.5	63.3 ± 12.8	69.1 ± 12.3	65.9 ± 11.9	<0.001
Hypertension	789,126 (57.8%)	161,354 (53.9%)	292,946 (58.5%)	317,043 (60.4%)	1,388,315 (57.1%)	325,689 (52.0%)	487,063 (58.7%)	551,155 (60.1%)	<0.001
Dyslipidemia	535,093 (39.2%)	114,278 (38.2%)	187,976 (37.5%)	225,695 (43.0%)	1,116,106 (45.9%)	283,010 (45.2%)	375,542 (45.2%)	444,037 (48.4%)	<0.001
Diabetes	406,473 (29.8%)	84,321 (28.2%)	174,432 (34.8%)	135,568 (25.8%)	625,187 (25.7%)	132,554 (21.2%)	255,366 (30.8%)	223,238 (24.4%)	<0.001
PAD	53,290 (3.9%)	9814 (3.3%)	25,305 (5.1%)	16,744 (3.2%)	128,423 (5.3%)	21,672 (3.5%)	58,384 (7.0%)	45,563 (5.0%)	<0.001
Prior Stroke	13,499 (1.0%)	4706 (1.6%)	6517 (1.3%)	1546 (0.3%)	17,754 (0.7%)	6144 (1.0%)	8220 (1.0%)	2640 (0.3%)	<0.001
Atrial Fibrillation	264,386 (19.4%)	50,471 (16.9%)	128,443 (25.6%)	77,167 (14.7%)	373,779 (15.4%)	73,106 (11.7%)	170,200 (20.5%)	121,625 (13.3%)	<0.001
COPD	90,138 (6.6%)	15,335 (5.1%)	41,415 (8.3%)	30,932 (5.9%)	179,716 (7.4%)	31,396 (5.0%)	80,640 (9.7%)	63,208 (6.9%)	<0.001
Renal Insufficiency	277,492 (20.3%)	53,135 (17.8%)	139,090 (27.8%)	76,638 (14.6%)	420,295 (17.3%)	79,593 (12.7%)	198,771 (23.9%)	131,869 (14.4%)	<0.001
Prior CABG	63,026 (4.6%)	6626 (2.2%)	24,630 (4.9%)	30,665 (5.8%)	199,755 (8.2%)	21,223 (3.4%)	76,097 (9.2%)	98,911 (10.8%)	<0.001
*During hospital stay*									
Mean Hospital Stay Duration, Days	7.7 ± 7.8	8.9 ± 8.6	9.2 ± 8.6	5.5 ± 5.5	6.9 ± 7.6	8.2 ± 8.2	8.2 ± 8.3	4.9 ± 5.8	<0.001
Coronary Angiography	754,492 (55.2%)	192,487 (64.3%)	261,139 (52.1%)	294,048 (56.0%)	1,608,615 (66.2%)	470,259 (75.1%)	541,030 (65.2%)	582,325 (63.5%)	<0.001
Inhospital PCI	472,421 (34.6%)	171,602 (57.3%)	166,179 (33.2%)	130,142 (24.8%)	1,136,066 (46.7%)	433,077 (69.2%)	379,946 (45.8%)	312,110 (34.1%)	<0.001
Inhospital CABG	43,408 (3.2%)	9116 (3.1%)	18,909 (3.8%)	14,266 (2.7%)	136,341 (5.6%)	29,327 (4.7%)	58,956 (7.1%)	44,653 (4.9%)	<0.001
Inhospital Mortality	107,647 (7.9%)	49,122 (16.4%)	40,484 (8.1%)	3297 (0.6%)	132,311 (5.4%)	61,968 (9.9%)	47,821 (5.8%)	4962 (0.5%)	<0.001

**Table 2 jcm-09-02124-t002:** Characteristics of female and male patients with ACS between 2005 and 2015; see Appendix A for results displayed by year.

	Females		Males		*p*-Value—2005	*p*-Value—2015
	2005	2015	2005	2015		
N available	140,755	108,292	232,444	209,146	<0.001	<0.001
Age (years)	72.7 ± 11.9	73.2 ± 12.4	65.6 ± 12.2	67.0 ± 12.7	<0.001	<0.001
Hypertension	80,346 (57.1%)	63,780 (58.9%)	130,153 (56.0%)	121,749 (58.2%)	<0.001	0.001
Dyslipidemia	50,992 (36.2%)	45,576 (42.1%)	100,662 (43.3%)	101,255 (48.4%)	<0.001	<0.001
Diabetes	40,858 (29.0%)	32,068 (29.6%)	53,755 (23.1%)	57,241 (27.4%)	<0.001	<0.001
PAD	4757 (3.4%)	4929 (4.6%)	11,218 (4.8%)	12,365 (5.9%)	<0.001	<0.001
Prior stroke	1656 (1.2%)	964 (0.9%)	1987 (0.9%)	1506 (0.7%)	<0.001	<0.001
Atrial fibrillation	24,353 (17.3%)	22,832 (21.1%)	30,371 (13.1%)	36,826 (17.6%)	<0.001	<0.001
COPD	8376 (6.0%)	8087 (7.5%)	16,840 (7.2%)	15,910 (7.6%)	<0.001	0.371
Renal insufficiency	21,728 (15.4%)	25,858 (23.9%)	33,080 (14.2%)	39,229 (18.8%)	<0.001	<0.001
Prior CABG	5845 (4.2%)	5186 (4.8%)	18,183 (7.8%)	17,347 (8.3%)	<0.001	<0.001
*During hospital stay*						
Mean hospital-stay duration (days)	8.4 ± 8.0	6.8 ± 7.0	7.4 ± 7.5	6.4 ± 6.9	<0.001	<0.001
Coronary angiography	63,248 (44.9%)	69,879 (64.6%)	132,468 (57.0%)	153,857 (73.6%)	<0.001	<0.001
Inhospital PCI	37,193 (26.4%)	46,276 (42.7%)	89,074 (38.3%)	113,369 (54.2%)	<0.001	<0.001
Inhospital CABG	4588 (3.3%)	3336 (3.1%)	13,115 (5.6%)	11,547 (5.5%)	<0.001	<0.001
Inhospital mortality	11,582 (8.2%)	7766 (7.2%)	12,450 (5.4%)	11,064 (5.3%)	<0.001	<0.001

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
