# Peer review of "Sex-Specific Outcomes in Patients with Acute Coronary Syndrome"

_jcm, 2020, doi:10.3390/jcm9072124_

Round 1

Reviewer 1 Report

This study retrospectively sought investigate sex-specific changes of incidence, outcome and related interventions in patients diagnosed with ACS in the past decade in Germany.

In total, 1,366,045 females and 2,431,501 males presenting with ACS were recorded.  They concluded that they report sex differences in incidence, treatment and outcome of ACS patients within the past decade in Germany. Women had a substantially higher mortality and a lower rate of coronary interventions.

This is an interesting and clinically relevant study. The strengths of this study are the large number of patients with ACS who were analyzed and the magnitude of detail of the analysis.

My comments are related to the following points:

1) The authors conclude that women had a substantially higher mortality and a lower rate of coronary interventions. This result by itself has previously been reported several times. What appears to be unique about this study? In this regard, the novel information provided by the present study with regard to existing knowledge should be further clarified.

2) In subgroup analyses, the incidence rate of UAP and STEMI decreased as well in both sexes, but the incidence rate of NSTEMI increased from 1.01 cases/1,000 person-years to 1.34 cases/1,000 person-years in females and even stronger in males from 1.67 cases/1,000 person-years to 2.54 cases/1,000 person-years. The authors should discuss why the incidence rate of NSTEMI increased.

3) In both sexes, there was a strong decrease of in-hospital mortality from 2005 to 2015. Which factors were associated with a strong decrease of in-hospital mortality?

Author Response

Reviewer #1

This is an interesting and clinically relevant study. The strengths of this study are the large number of patients with ACS who were analyzed and the magnitude of detail of the analysis.

My comments are related to the following points:

  1. The authors conclude that women had a substantially higher mortality and a lower rate of coronary interventions. This result by itself has previously been reported several times. What appears to be unique about this study? In this regard, the novel information provided by the present study with regard to existing knowledge should be further clarified.

Answer: We thank the reviewer for the helpful and critical review of the submitted manuscript. We agree, that earlier studies reported important gender differences in patients with myocardial infarction. We believe, that our findings add to this prior evidence due to the contemporary nature of our data set being representative of current ACS treatments and due to the very large sample size capturing all events in Germany in one decade. In the revised discussion section, we do now emphasize these aspects.

  1. In subgroup analyses, the incidence rate of UAP and STEMI decreased as well in both sexes, but the incidence rate of NSTEMI increased from 1.01 cases/1,000 person-years to 1.34 cases/1,000 person-years in females and even stronger in males from 1.67 cases/1,000 person-years to 2.54 cases/1,000 person-years. The authors should discuss why the incidence rate of NSTEMI increased.

Answer: As suggested by the reviewer, we do now address this topic in the revised discussion section on page 8:

Recent literature reported, as did our study, a decrease of ACS incidence over time. In contrast to that, the proportion of NSTEMI cases increased. Reasons for this increase might be a better detection of even small scale myocardial injury using high-sensitivity troponin assays, as well as an aging population with more frequent comorbidities.(1, 2)

  1. In both sexes, there was a strong decrease of in-hospital mortality from 2005 to 2015. Which factors were associated with a strong decrease of in-hospital mortality?

Answer: We fully agree, that this is an important question, which would help to interpret the change in mortality. Unfortunately, due to the study design, it is not possible to perform regression analyses in the present data set. The Research Data Center of the Federal Bureau of Statistics and Statistical Offices of the Federal States provided aggregated statistics based on the requested ICD and OPS codes and based on the statistics, no regression analyses were possible. Thus, we are limited to the descriptive statistics.

Reviewer 2 Report

In the present manuscript, Neumann et al. show that short-term ACS outcomes, as measured by age-adjusted in-hospital mortality, are higher in women than in men. Conversely, age-adjusted incidence rates were substantially higher in men, as compared to women. Although similar findings have previously been reported, evidence from larger cohorts is scarce. The present registry encompasses a total of 3'797'546 ACS patients and is therefore one of the largest studies reported so far. For potential improvement, the authors might want to consider the following reviewer comments:

  • Women are historically underrepresented in ACS clinical trials, while underlying causes are controversially discussed. Please discuss the potential implications of female underrepresentation in ACS studies, and whether this may contribute to the excess post-MI mortality observed in the present study and other reports consistent with the present results.

  • Among the reported risk factors, smoking and family history of CAD are missing. Please mention this in the text and discuss how they differ between sexes based on the available body of literature.

  • Throughout the manuscript, the authors do not provide any P values. When comparing mean values between women and men, it is important that the P value is provided to support the claim of significant sex differences. Did the authors perform any post hoc analysis following one-way ANOVA? Please provide the significance values for group comparisons in tables 1 and 2. It is not clear, for instance, what baseline characteristics were statistically different between women and men.

  • As correctly stated by the authors, previous studies have shown that sex differences in short-term mortality disappeared upon age-adjustment, particularly in elderly patient populations. In the present study, the authors found that sex differences were robust and persisted after adjusting for age. What might be possible explanations for these seemingly controversial findings?

  • Recent studies demonstrated that sex differences in ACS mortality are particularly relevant in younger populations. Younger women seemed to have a significantly higher ACS mortality rates than age-matched men [1]. It is a pity that the authors do not present age-stratified data ( young vs. elderly women and young vs. elderly men, respectively). It might be helpful to unravel the underlying cause for above described “controversial findings”.

  • Among the potential mechanisms for the higher ACS mortality women, this reviewer suggests that the development of non-obstructive CAD and microvascular angina should be discussed in more detail. Visually non-detectable microvascular angina is not a benign condition, and its diagnostic workup is rather complex, as compared to epicardial stenosis. Hence, the higher prevalence of microvascular angina in female patients might, at least in part, explain the relatively high ACS mortality rates in women.

  • Please discuss the potential implications of non-traditional risk factors such as socioeconomic factors, system and patient delays and psychosocial burden within the context of this study.

In conclusion, this study provides solid evidence for the sex differences in short-term ACS outcomes due to the high sample size.

References

  • Haider A, Bengs S, Luu J, Osto E, Siller-Matula JM, Muka T et al. Sex and gender in cardiovascular medicine: presentation and outcomes of acute coronary syndrome. European Heart Journal. 2019;41(13):1328-36. doi:10.1093/eurheartj/ehz898.

Author Response

Reviewer #2

In the present manuscript, Neumann et al. show that short-term ACS outcomes, as measured by age-adjusted in-hospital mortality, are higher in women than in men. Conversely, age-adjusted incidence rates were substantially higher in men, as compared to women. Although similar findings have previously been reported, evidence from larger cohorts is scarce. The present registry encompasses a total of 3'797'546 ACS patients and is therefore one of the largest studies reported so far. For potential improvement, the authors might want to consider the following reviewer comments:

  1. Women are historically underrepresented in ACS clinical trials, while underlying causes are controversially discussed. Please discuss the potential implications of female underrepresentation in ACS studies, and whether this may contribute to the excess post-MI mortality observed in the present study and other reports consistent with the present results.

Answer: We thank the reviewer for this comment and have adapted the discussion section on page 9 accordingly:

In prior clinical trials investigating ACS patients, females were often at lower risk of adverse events, which might be explained by a certain underrepresentation.(3) As an example, in a large pooled analyses of trials by the TIMI group, only 29% of all participants were females. Thus, our data set capturing all ACS cases in Germany adds important knowledge to this possible selection bias.”

  1. Among the reported risk factors, smoking and family history of CAD are missing. Please mention this in the text and discuss how they differ between sexes based on the available body of literature.

Answer: We thank the reviewer for highlighting these missing risk factors, which were not captured in this data set. We have now adapted the limitations section on page 10:

“In addition, our understanding of the observed sex differences could have been improved by the knowledge of possible confounders such as medical treatment, cardiac biomarkers or other risk factors, such as smoking, family history or socioeconomic factors, that were unfortunately not available in the present data set.”

  1. Throughout the manuscript, the authors do not provide any P values. When comparing mean values between women and men, it is important that the P value is provided to support the claim of significant sex differences. Did the authors perform any post hoc analysis following one-way ANOVA? Please provide the significance values for group comparisons in tables 1 and 2. It is not clear, for instance, what baseline characteristics were statistically different between women and men.

Answer: As suggested by the reviewer, we have now included p-values for table 1 comparing the ACS population for females and males. Due to the large sample size all p-values were < 0.001. In subgroups comparing only UAP, STEMI or NSTEMI per sex, again the vast majority of p-values was < 0.001. For table 2 we include now include p-values comparing both sexes in 2005 and 2015. Again, nearly all p-values were < 0.001 (except for hypertension and COPD in 2015). The tables have been adapted accordingly.

  1. As correctly stated by the authors, previous studies have shown that sex differences in short-term mortality disappeared upon age-adjustment, particularly in elderly patient populations. In the present study, the authors found that sex differences were robust and persisted after adjusting for age. What might be possible explanations for these seemingly controversial findings?

Answer: We agree with the reviewer, that this is a controversial topic with diverging data. Indeed there is also data supporting a higher mortality for females during long-time follow-up. (4) Adjustment for risk factors and possible confounders might impact our findings. However, due to the nature of study design, these regression analyses were not possible. Thus, the interpretation of differences remains speculative. In the discussion section we have tried to capture the areas of differences in symptoms, treatment and risk factor distribution.

  1. Recent studies demonstrated that sex differences in ACS mortality are particularly relevant in younger populations. Younger women seemed to have a significantly higher ACS mortality rates than age-matched men [1]. It is a pity that the authors do not present age-stratified data ( young vs. elderly women and young vs. elderly men, respectively). It might be helpful to unravel the underlying cause for above described “controversial findings”.

Answer: We do thank the reviewer for highlighing this important topic. We have now included the crude rates of ACS rates and in-hospital mortality per age categories and for females and males in Table S6 (please see below). Interestingly, ACS cases occurred more often in younger males, compared to females, but the in-hospital mortality was slightly higher in females (e.g. aged 30 to < 40 years 4.11% vs. 8.95% of all ACS cases, while mortality 2.18% in females and 1.66% in males).

Table S6: Crude rates of ACS and in-hospital mortality per age categories for females and males.

Females

Males

Age categories

ACS

In-hospital mortality

ACS

In-hospital mortality

18 to <30

1,582 (0.12 %)

38 (2.4 %)

5,560 (0.23 %)

68 (1.22 %)

≥30 to <40

9,120 (0.67 %)

199 (2.18 %)

36,334 (1.49 %)

602 (1.66 %)

≥40 to <50

56,136 (4.11 %)

1,190 (2.12 %)

217,533 (8.95 %)

4,182 (1.92 %)

≥50 to <60

134,515 (9.85 %)

3,074 (2.29 %)

464,968 (19.12 %)

11,963 (2.57 %)

≥60 to <70

246,496 (18.04 %)

8,459 (3.43 %)

608,440 (25.02 %)

23,889 (3.93 %)

≥70 to <80

454,894 (33.3 %)

27,247 (5.99 %)

732,886 (30.14 %)

45,922 (6.27 %)

≥80 to <85

231,320 (16.93 %)

24,991 (10.8 %)

228,421 (9.39 %)

23,328 (10.21 %)

≥85 to <90

158,252 (11.58 %)

24,966 (15.78 %)

105,143 (4.32 %)

15,207 (14.46 %)

≥90

72,653 (5.32 %)

17,129 (23.58 %)

31,768 (1.31 %)

7,040 (22.16 %)

  1. Among the potential mechanisms for the higher ACS mortality women, this reviewer suggests that the development of non-obstructive CAD and microvascular angina should be discussed in more detail. Visually non-detectable microvascular angina is not a benign condition, and its diagnostic workup is rather complex, as compared to epicardial stenosis. Hence, the higher prevalence of microvascular angina in female patients might, at least in part, explain the relatively high ACS mortality rates in women.

Answer: As suggested by the reviewer, we have now included a short paragraph on this topic in the dicussion section:

A higher rate of non-significant coronary artery disease or microvascular disease could explain the lower PCI rates in our study, although this cannot be proven by our results. A potentially higher prevalence of small vessel disease in females could also be speculated as a reason for higher intervention rates in males compared to females.”

  1. Please discuss the potential implications of non-traditional risk factors such as socioeconomic factors, system and patient delays and psychosocial burden within the context of this study.

Answer: We thank the reviewer for for highlighing this important limitation. We have now adapted the limitations section on page 10:

“In addition, our understanding of the observed sex differences could have been improved by the knowledge of possible confounders such as medical treatment, cardiac biomarkers or other risk factors, such as smoking, family history or socioeconomic factors, that were unfortunately not available in the present data set.”

  1. In conclusion, this study provides solid evidence for the sex differences in short-term ACS outcomes due to the high sample size.

Answer: We thank the reviewer for taking the time to critically review the manuscript.

Round 2

Reviewer 2 Report

The authors have addressed all comments properly.